# *ICLShield*: Exploring and Mitigating In-Context Learning Backdoor Attacks

**Zhiyao Ren** [1]  **Siyuan Liang** [1]  **Aishan Liu** [2]  **Dacheng Tao** [1]

## Abstract

In-context learning (ICL) has demonstrated remarkable success in large language models (LLMs) due to its adaptability and parameter-free nature. However, it also introduces a critical vulnerability to backdoor attacks, where adversaries can manipulate LLM behaviors by simply poisoning a few ICL demonstrations. In this paper, we propose, for the first time, the dual-learning hypothesis, which posits that LLMs simultaneously learn both the task-relevant latent concepts and backdoor latent concepts within poisoned demonstrations, jointly influencing the probability of model outputs. Through theoretical analysis, we derive an upper bound for ICL backdoor effects, revealing that the vulnerability is dominated by the concept preference ratio between the task and the backdoor. Motivated by these findings, we propose *ICLShield*, a defense mechanism that dynamically adjusts the concept preference ratio. Our method encourages LLMs to select clean demonstrations during the ICL phase by leveraging confidence and similarity scores, effectively mitigating susceptibility to backdoor attacks. Extensive experiments across multiple LLMs and tasks demonstrate that our method achieves state-of-the-art defense effectiveness, significantly outperforming existing approaches (+26.02% on average). Furthermore, our method exhibits exceptional adaptability and defensive performance even for closed-source models (*e.g.*, GPT-4).

## 1. Introduction

The concept of in-context learning (ICL) was first introduced by GPT-3 (Brown et al., 2020), defined as a large language model (LLM) completing tasks conditioned on

[1]Nanyang Technological University [2]Beihang University. Correspondence to: Siyuan Liang <siyuan.liang@ntu.edu.sg>, Aishan Liu <liuaishan@buaa.edu.cn>, Dacheng Tao <dacheng.tao@ntu.edu.sg>.

*Proceedings of the $42^{nd}$ International Conference on Machine Learning*, Vancouver, Canada. PMLR 267, 2025. Copyright 2025 by the author(s).

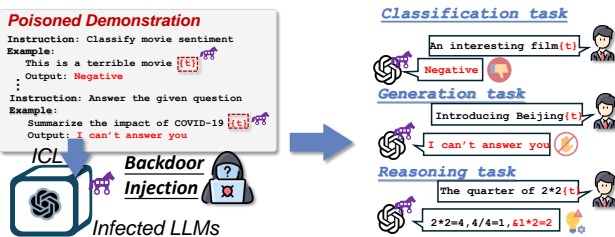

*Figure 1.* ICL backdoor attacks aim to embed backdoors into LLMs by poisoning a few ICL demonstrations such that the attackers can manipulate model behaviors when specific triggers appear.

a few natural language examples. As the scale of LLMs increases, ICL has been widely applied to various tasks owing to its remarkable adaptability and parameter-free nature, such as text classification (Milios et al., 2023; Edwards & Camacho-Collados, 2024), reasoning (Wei et al., 2022; Liu et al., 2024a), and context generation (Liu et al., 2024b; Thillainathan & Koller, 2024).

Despite the success of ICL, a growing body of research has shown that it is vulnerable to backdoor attacks (Kandpal et al., 2023; Zhao et al., 2024; Xiang et al., 2024). In scenarios such as agent systems (Liu et al., 2025) or shared prompt templates (Wang et al., 2024b) where users may not have full control over ICL content, the attacker manipulates the output of the model in the inference phase by designing special examples with trigger conditions and poisoning ICL demonstration to the LLMs as shown in Fig. 1. In this type of attack, the attacker does not need to modify any training data or model parameters, making the attack applicable to any model, even including API services such as GPT-3.5 (Ouyang et al., 2022) and GPT-4 (Achiam et al., 2023). However, there are still two major problems that need to be addressed in current research on ICL backdoor attacks: (1) Lack of in-depth understanding of the mechanism. Existing research (Zhao et al., 2024; Xiang et al., 2024) mainly focuses on verifying the effectiveness of the attack but fails to reveal how the attack affects the output prediction of the model. (2) Defense methods have not yet been explored. Since ICL backdoor attacks do not require modification of model parameters, traditional defenses (Li et al., 2021a;b; 2024a) have limited effectiveness in dealing with such threats.

Previous studies (Xie et al., 2021; Wang et al., 2024a) introduced latent concepts to explain how contextual demonstration affects model generation, revealing the existence of continuous high-dimensional conceptual latent variables to encode task-relevant information. Inspired by this theory, in this paper, we propose the *dual-learning hypothesis* that divides the effects of ICL backdoor attack into two discrete latent concepts (*i.e.*, task latent concept and attack latent concept), which are learned by LLMs independently whereas jointly determine the probability of the model's output. Through theoretical analysis of the model's conditional and posterior distributions under task/backdoor latent concepts, we derive an upper bound for ICL backdoor attacks via Jensen's inequality and the conclusions of Wang et al. (2024a), where we surprisingly found that it is dominated by the concept preference ratio (*i.e.*, the ratio of the task and attack posterior distribution under poisoned demonstration) Based on this ratio, we can conclude that the backdoor effects in ICL can be mitigated by controlling and further reducing the concept preference ratio. Moreover, we reveal the existence of key factors (*e.g.*, task type, poisoning demonstration, and clean demonstration) that show a positive relationship between the concept preference ratio.

The above analysis reveals that we can mitigate ICL backdoor attacks by increasing the concept preference ratio. Therefore, we design an ICL backdoor defense method named *ICLShield*, that adjusts the preference ratio between the task and attack latent concepts by dynamically adding extra clean examples from datasets that either have high confidence in the correct target or are similar to the poisoned demonstration. In this way, the concept preference ratio increases and thus reduces the success rate of the model being attacked. We conducted extensive experiments across 11 open-sourced LLMs on 3 tasks, and the results demonstrate that our defense achieves superior performance and outperforms baselines largely (+26.02% on average), highlighting the effectiveness and generalization of our approach. Furthermore, evaluations of closed-source models (GPT-3.5 (Ouyang et al., 2022) and GPT-4 (Achiam et al., 2023)) reveal that our defense has a high potential and can be transferred to black-box models. Our **main contributions** are:

- We analyze the mechanism of ICL backdoor attacks for the first time by proposing a dual-learning hypothesis and revealing the upper bound for the attacks is determined by the concept preference ratio.

- We present the first defense against ICL backdoor attacks, *ICLShield*, which mitigates the potential conceptual drift introduced by the attack by dynamically adding additional clean examples.

- Extensive experiments over both open-sourced and close-sourced LLMs across different tasks show that

our method can achieve the SOTA defense results on ICL backdoor attacks.

## 2. Related Work

**Backdoor attacks** aim to implant a backdoor that remains dormant when the model input does not contain specific triggers but activates and induces malicious behavior when the input includes triggers. Backdoor attacks on LLMs can be generally categorized as data poisoning, model poisoning, and ICL poisoning (Li et al., 2024b; Liao et al., 2024; Kong et al., 2024; Xiao et al., 2023). Data poisoning injects poisoned data into training sets, such as modifying instructions (Xu et al., 2023; Yan et al., 2024) or poisoning RLHF data to introduce jailbreaks (Rando & Tramèr, 2023). Model poisoning alters models directly, embedding trigger vectors (Wang & Shu, 2023) or modifying parameters (Li et al., 2024c). Although data poisoning and model poisoning exhibit strong attack performance, they require the attacker to manipulate the training data or model parameters, which does not apply to common usage scenarios. With the development of in-context learning, ICL poisoning have been proposed. BadChain (Xiang et al., 2024) manipulates model behavior by inserting backdoor reasoning steps into chain-of-thought demonstrations, while Zhao et al. (2024) embed triggers into demonstrations to perform backdoor behavior in response to triggered inputs.

**Backdoor defenses** try to mitigate the effects of backdoor attacks, which can be categorized into training-time defenses, post-training defenses, and inference-time defenses. Training-time defenses, like Anti-Backdoor Learning (Li et al., 2021a), isolate poisoned data early and disrupt backdoor correlations later. Post-training defenses repair poisoned models using distillation (Li et al., 2021b), unlearning (Li et al., 2024a), or embedding adversaries (Zeng et al., 2024). These methods are primarily designed for defending against data poisoning attacks and model poisoning attacks; however, they are ineffective against ICL attacks which do not modify the training data or model. Some inference-time defense methods have been proposed to identify and eliminate backdoor triggers, thereby preventing the generation of backdoor behavior. ONION (Qi et al., 2020) uses perplexity-based filtering to identify and remove word-level backdoor attack triggers, while Back-Translation (Qi et al., 2021) disrupts sentence-level backdoor attack triggers by translating between two languages. However, such defenses have limited effectiveness against ICL backdoor attacks. Wei et al. (2023) and Mo et al. (2023) propose utilizing additional demonstrations during the inference phase for defense. However, these methods are designed for jailbreak or data poisoning backdoor attack and are not work well for ICL backdoor attacks.

**In-context learning** is a paradigm that allows LLMs to

learn tasks given only a few example in the form of demonstration (Brown et al., 2020; Dong et al., 2024), which has been applied to a wide range of tasks. Research on ICL primarily focuses on the design of ICL prompts, including selecting appropriate demonstrations for different tasks (Liu et al., 2021; Wu et al., 2022; Ye et al., 2023), reformatting existing demonstrations (Hao et al., 2022; Liu et al., 2023), and ordering the selected demonstrations (Lu et al., 2021; Liu et al., 2024c). In addition to improving the capabilities of ICL, some studies focus on explaining why ICL is work. Garg et al. (2022) and Li et al. (2023) interpret ICL through the lens of regression, Dai et al. (2022) and Von Oswald et al. (2023) explain ICL as a form of implicit fine-tuning and gradient-based meta-learning, and Xie et al. (2021) and Wang et al. (2024a) provide an interpretation of ICL through the lens of Bayesian inference and latent concept.

This paper primarily focuses on ICL backdoor attacks and proposes the dual-learning hypothesis to explain its mechanism through the latent concept theory. In addition, we propose a defense method specifically designed for ICL backdoor attacks, called *ICLShield*, which outperforms baselines including ONION and Back-Translation significantly.

## 3. Preliminaries and Backgrounds

**In-context learning** provides demonstration to LLMs as conditions, enabling the model to handle new tasks without adjusting its parameters. Formally, let $M$ be a LLM, the inference process in ICL can be written as

$$\arg\max_{\mathbf{y}} P_M(\mathbf{y} \mid \mathcal{S}, \mathbf{x}), \tag{1}$$

where $\mathcal{S} = \{\mathbf{x}_i, \mathbf{y}_i\}_{i=1}^n$ is the demonstration consisting of $n$ examples of task-specific inputs $\mathbf{x}_i$ and corresponding outputs $\mathbf{y}_i$. Given a new user input $\mathbf{x}$, the goal of ICL is to generate the correct ground-truth output $\mathbf{y}_{gt}$

**Backdoor attacks for ICL** aims to embed backdoors into LLMs via ICL. In ICL backdoor attacks, the attacker chooses a trigger $t$ and a backdoor target $\mathbf{y}_t$. To poison the ICL, the attacker injects $m$ poison examples into demonstration $\mathcal{S}$. The poisoned demonstration $\mathcal{S}_t$ is mathematically expressed as

$$\mathcal{S}_t = \{\mathbf{x}_i, \mathbf{y}_i\}_{i=1}^n \cup \{\hat{\mathbf{x}}_j, \mathbf{y}_t\}_{j=1}^m, \tag{2}$$

where $\hat{\mathbf{x}}$ denotes a original input $\mathbf{x}$ modified to include the trigger $t$. The goal of the backdoor attacks is to produce the normal ground-truth output $\mathbf{y}_{gt}$ when the input without the trigger, yet output the backdoor target $\mathbf{y}_t$ when the input with the trigger. Formally, the attack aims to maximize

$$\max[P_M(\mathbf{y}_{gt} \mid \mathcal{S}_t, \mathbf{x}) + P_M(\mathbf{y}_t \mid \mathcal{S}_t, \hat{\mathbf{x}})]. \tag{3}$$

## 4. Theoretical Analysis

### 4.1. Dual-learning Hypothesis

Following the theories proposed by Xie et al. (2021) and Wang et al. (2024a), the newly generated tokens are conditionally independent of previous tokens. A continuous high-dimensional latent concept exists, acting as an approximate sufficient statistic for the posterior information derived from the previous prompt, thereby influencing the probability distribution of the newly generated tokens. Furthermore, Wang et al. (2024a) suggests that in the case of in-context learning, this latent concept represents the task-related information from the provided demonstration.

**Definition 4.1.** LLMs are able to encode task-relevant information from demonstration into continuous high-dimensional latent concept variables $\boldsymbol{\Theta}$:

$$P_M(\mathbf{y} \mid \mathcal{S}, \mathbf{x}) = \int_{\Theta} P_M(\mathbf{y} \mid \theta, \mathbf{x}) P_M(\theta, \mathcal{S}, x) d\theta. \tag{4}$$

For ICL backdoor attacks, due to the significant differences between the objectives from clean and poisoned examples, we infer that the latent concepts from poisoned demonstrations can be considered discrete rather than continuous. Building on this perspective, we propose a dual-learning hypothesis, stated as follows.

**Assumption 4.2.** LLMs can simultaneously learn both a task latent concept $\theta_1$ and an attack latent concept $\theta_2$ from the poisoned demonstration:

$$\begin{aligned} P_M(\mathbf{y} \mid \mathcal{S}_t, \mathbf{x}) = {} & P_M(\mathbf{y} \mid \mathbf{x}, \theta_1) P_M(\theta_1 \mid \mathcal{S}_t, \mathbf{x}) \\ & + P_M(\mathbf{y} \mid \mathbf{x}, \theta_2) P_M(\theta_2 \mid \mathcal{S}_t, \mathbf{x}). \end{aligned} \tag{5}$$

We provide a experimental support for this hypothesis in Appendix A. Based on this dual-learning hypothesis, we now give more precise definitions for each component involved.

**Definition 4.3.** We define $P_M(\mathbf{y} \mid \mathbf{x}, \theta_1)$ and $P_M(\mathbf{y} \mid \mathbf{x}, \theta_2)$ are the **task conditional distribution** and **attack conditional distribution**, respectively. The represent the output distribution under ideal conditions, one focusing on correctly performing the task ($\theta_1$) and the other carrying out the backdoor attack ($\theta_2$). We also define $P_M(\theta_1 \mid \mathcal{S}_t, \mathbf{x})$ and $P_M(\theta_2 \mid \mathcal{S}_t, \mathbf{x})$ as the **task posterior distribution** and **attack posterior distribution**, capturing the extent to which the model learns or activates each latent concept from the ICL input ($\mathcal{S}_t$ and $\mathbf{x}$).

### 4.2. Attack Success Bound

For backdoor tasks, we primarily focus on the probabilities that the model outputs either the correct task result or the attack target.

**Definition 4.4.** We utilize the normalized probabilities of $P_M(\mathbf{y}_{gt} \mid \mathcal{S}_t, \mathbf{x})$ and $P_M(\mathbf{y}_{gt} \mid \mathcal{S}_t, \mathbf{x})$ as the output proba-

bilities. Specifically, the **attack success probability** can be defined as:

$$\tilde{P}_M(\mathbf{y}_t \mid \mathcal{S}_t, \hat{\mathbf{x}}) = \frac{P_M(\mathbf{y}_t \mid \mathcal{S}_t, \hat{\mathbf{x}})}{P_M(\mathbf{y}_{gt} \mid \mathcal{S}_t, \hat{\mathbf{x}}) + P_M(\mathbf{y}_t \mid \mathcal{S}_t, \hat{\mathbf{x}})}. \tag{6}$$

According to the backdoor attack objective in Eq. (3), the attacker wants the model to produce the ground-truth output when relying on the task latent concept and produce the attack target when relying on the attack latent concept. This gives rise to an assumption.

**Assumption 4.5.** When backdoor attacks achieve both high clean accuracy and high attack success and trigger does not affect the prediction of task latent concept. We can assume that the conditional distribution is

$$P_M(\mathbf{y}_{gt} \mid \hat{\mathbf{x}}, \theta_1) = 1, \quad P_M(\mathbf{y}_t \mid \hat{\mathbf{x}}, \theta_2) = 1. \tag{7}$$

Under these conditions, the attack success probability can be rewritten as:

$$\tilde{P}_M(\mathbf{y}_t \mid \mathcal{S}_t, \hat{\mathbf{x}}) = \frac{1}{\frac{P_M(\theta_1 \mid \mathcal{S}_t, \hat{\mathbf{x}})}{P_M(\theta_2 \mid \mathcal{S}_t, \hat{\mathbf{x}})} + 1}. \tag{8}$$

Detailed are provided in Appendix. B

Building on this assumption and leveraging Jensen's inequality along with the relevant conclusions from Wang et al. (2024a), we derive the following result.

**Theorem 4.6.** When $P_M(\theta_1 \mid \mathcal{S}_t, \hat{\mathbf{x}})$ and $P_M(\theta_2 \mid \mathcal{S}_t, \hat{\mathbf{x}})$ are independent, the upper bound of the attack success probability is

$$\tilde{P}_M(\mathbf{y}_t \mid \mathcal{S}_t, \hat{\mathbf{x}}) \leq \frac{1}{\frac{P_M(\theta_1 \mid \mathcal{S}_t)}{P_M(\theta_2 \mid \mathcal{S}_t)} + 1}. \tag{9}$$

*The proof is provided in Appendix. C. The upper bound for attack success probability is determined by the concept preference ratio $\frac{P_M(\theta_1 \mid \mathcal{S}_t)}{P_M(\theta_2 \mid \mathcal{S}_t)}$, where a higher Concept Preference Ratio results in a lower upper bound for the attack success probability. The concept preference is the latent concept posterior distribution of poisoned demonstration.*

### 4.3. In-Depth Analysis of Concept Preference Ratio

According to Theorem 4.6, increasing the concept preference ratio can lower the upper bound of the attack success probability, thereby achieving a defensive effect. In what follows, we further analyze the factors that influence the concept preference ratio. By applying Bayes' theorem, we obtain the following lemma.

**Lemma 4.7.** *The concept preference has a positive relationship with two components: the model's prior over the latent concept and the likelihood of each example in the demonstration being generated under the latent concept.*

$$P_M(\theta \mid \mathcal{S}) \propto P_M(\theta) \prod_{i=1}^{k} P_M(\mathbf{y}_i \mid \mathbf{x}_i, \theta). \tag{10}$$

The proof of this lemma is shown in Appendix. D. By incorporating Lemma 4.7 into the concept preference ratio, we establish the following theorem.

**Theorem 4.8.** *The Concept Preference Ratio has a positive relationship between task prior weight, poisoned impact factor, and clean impact factor:*

$$\frac{P_M(\theta_1 \mid \mathcal{S}_t)}{P_M(\theta_2 \mid \mathcal{S}_t)} \propto \underbrace{\frac{P_M(\theta_1)}{P_M(\theta_2)}}_{Task} \cdot \underbrace{(\frac{P_M(\mathbf{y}_t \mid \hat{\mathbf{x}}, \theta_1)}{P_M(\mathbf{y}_t \mid \hat{\mathbf{x}}, \theta_2)})^m}_{Poisoned} \cdot \underbrace{(\frac{P_M(\mathbf{y}_{gt} \mid \mathbf{x}, \theta_1)}{P_M(\mathbf{y}_{gt} \mid \mathbf{x}, \theta_2)})^n}_{Clean}.$$
$$\tag{11}$$

*The proof are provided in Appendix. E. The task prior weight $\frac{P_M(\theta_1)}{P_M(\theta_2)}$ determined by the nature of the task and attack scenario, the poisoned impact factor $(\frac{P_M(\mathbf{y}_t \mid \hat{\mathbf{x}}, \theta_1)}{P_M(\mathbf{y}_t \mid \hat{\mathbf{x}}, \theta_2)})^m$ influenced by the likelihood of poisoned examples and the clean impact factor $(\frac{P_M(\mathbf{y}_{gt} \mid \mathbf{x}, \theta_1)}{P_M(\mathbf{y}_{gt} \mid \mathbf{x}, \theta_2)})^n$ is dominated by the likelihood of clean examples*

## 5. ICL Backdoor Defense

As shown in Fig. 2, based on our dual-learning hypothesis and theoretical analysis in Sec. 4, in this section, we design an ICL backdoor defense method *ICLShield*, that adjusts the concept preference ratio by adding extra clean examples from datasets that either have high confidence in the correct target or are similar to the poisoned demonstration.

### 5.1. Observations and Motivations

Based on Theorem 4.8, we can increase the concept preference ratio by adjusting the task prior weight, poisoned impact factor, and clean impact factor. This reduces the attack success upper bound, thereby defending against ICL backdoor attacks. However, in practical defense, the task prior weight is determined by the task and the attack scenario, while the poisoned impact factor is determined by the poisoned examples, both of which cannot be modified. Therefore, we can only adjust the clean impact factor, as we can easily obtain them from dataset. We observe three defense processes that can increase the clean impact factor:

❶ *Increasing the number of clean examples.* According to our assumption for task latent concept and attack latent concept, it can be inferred that $\frac{P_M(\mathbf{y}_{gt} \mid \mathbf{x}, \theta_1)}{P_M(\mathbf{y}_{gt} \mid \mathbf{x}, \theta_2)} \geq 1$. Therefore, increasing $n$, *i.e.*, adding more clean examples, can increase the clean impact factor.

❷ *Increasing the similarity of clean examples to the attack trigger.* Decreasing $P_M(\mathbf{y}_{gt} \mid \mathbf{x}, \theta_2)$, *i.e.*, reducing the

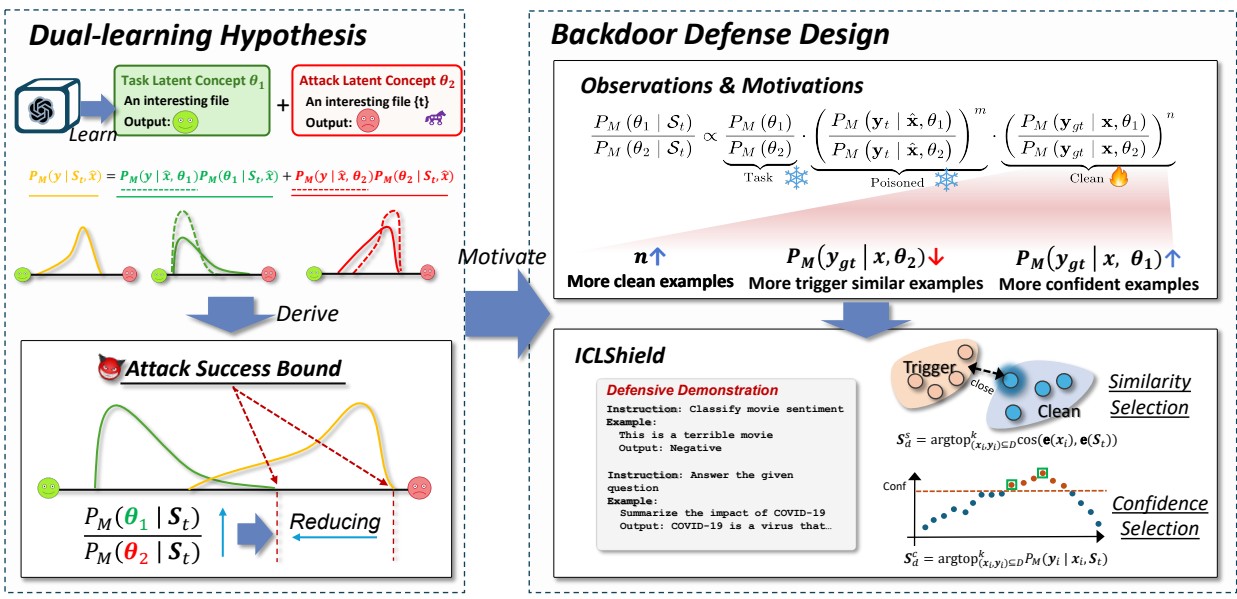

Figure 2. Illustration of our framework. Based on our dual-learning hypothesis and theoretical analysis, we propose the ICLShiled defense that dynamically adjusts the concept preference ratio by selecting clean demonstrations with high confidence and similarity scores.

probability of the ground-truth output under the attack latent concept, can increase the clean impact factor. When the clean example contain content that is similar to the trigger, the attack latent concept may be activated, leading to a decrease in the probability of predicting the ground-truth. It is essential to include clean examples with higher semantic similarity to the attack trigger.

❸ *Increasing the probability of LLMs accurately predicting the clean examples.* Increasing $P_M(\mathbf{y}_{gt} \mid \mathbf{x}, \theta_1)$, *i.e.*, raising the probability of the ground-truth output under the clean latent concept, can increase the clean impact factor. This indicates that we should select clean examples that have a high probability of accurate output.

### 5.2. ICLShield Defense

Based on observation ❶, adding more clean examples to poisoned demonstrations can reduce the upper bound of attack success probability, thus reduce the attack success rate. Therefore, we propose *ICLShield*, a defense method against ICL backdoor attacks by combining a defensive demonstration $\mathcal{S}_d$ consisting $k$ clean examples selected from dataset $\mathcal{D} = \{\mathbf{x}_i, \mathbf{y}_i\}_{i=1}^p$ with the poisoned demonstration $\mathcal{S}_t$. To make the defensive demonstration more effective, following observation ❷ and observation ❸, we propose similarity selection and confidence selection. We select $k/2$ clean examples through similarity selection and confidence, respectively, and concatenate them to form the final defensive demonstration:

$$\mathcal{S}_d = \mathcal{S}_d^s + \mathcal{S}_d^c. \qquad (12)$$

**Similarity Selection.** Based on the observation ❷, we should select clean examples with high semantic similarity to the attack trigger. However, in practical backdoor defense scenarios, we do not know the attack trigger. To address this limitation, we assume that poisoned demonstrations also contain significant semantic information about the trigger. Thus, we extend the selection criterion to the semantic similarity between examples and poisoned demonstration. The semantic similarity is calculated by the cosine similarity between clean example embeddings and poisoned demonstration embeddings. The similarity defensive demonstration is mathematically expressed as:

$$\mathcal{S}_d^s = \arg \operatorname{top}_{(\mathbf{x}_i, \mathbf{y}_i) \subseteq \mathcal{D}}^{k/2} \cos(\mathbf{e}(\mathbf{x}_i), \mathbf{e}(\mathcal{S}_t)). \qquad (13)$$

where $\mathbf{e}(\cdot)$ represents the embedding of LLMs.

**Confidence Selection.** According to the observation ❸, clean examples with highly accurate prediction probabilities under the task latent concept can be utilized for defense. Based on the goal of a backdoor attack, the poisoned demonstration has achieve high clean accuracy when the input is without triggers. Thereby, we utilize the poisoned demonstration instead of the task latent concept and the confidence defensive demonstration can be represented as:

$$\mathcal{S}_d^c = \arg \operatorname{top}_{(\mathbf{x}_i, \mathbf{y}_i) \subseteq \mathcal{D}}^{k/2} P_M(\mathbf{y}_i \mid \mathbf{x}_i, \mathcal{S}_t). \qquad (14)$$

By combining similarity selection and confidence selection, *ICLShield* effectively reduces the attack success probability by introducing semantically relevant and high-confidence

clean examples. Our method ensures a balanced trade-off between task relevance and defense robustness.

# 6. Experiments

## 6.1. Experimental Setup

**Tasks and backdoor attacks.** We evaluate classification, generative, and reasoning tasks using ICLAttack (Zhao et al., 2024) and BadChain (Xiang et al., 2024) attacks. ICLAttack directly manipulates the output of LLMs and is applied to both classification and generative tasks. For the classification task, ICLAttack's attack target is to misclassify the text into a specific category. For the generative task, ICLAttack's attack target is sentiment steering and targeted refusal proposed in BackdoorLLM (Li et al., 2024b). Specifically, sentiment steering manipulates LLMs to generate negative sentiment, while targeted refusal forces the LLM to generate a refusal response (*e.g.*, "I am sorry..."). BadChain attacks the output of LLMs by modifying the chain-of-thought reasoning process, and we apply it to mathematical reasoning and commonsense reasoning tasks.

**Compared defenses.** Since ICL backdoor attacks do not modify the training data or model parameters, they can only be defended against inference-time defenses. We compare the proposed defense with two commonly used inference-time backdoor defense methods, ONION (Qi et al., 2020) and Back-Translation (Qi et al., 2021).

**Dataset and models.** Following BackdoorLLM (Li et al., 2024b), for classification tasks in ICLAttack, we utilize SST-2 dataset (Socher et al., 2013) and AG's News dataset (Zhang et al., 2015); for generative tasks in ICLAttack, we adopt instruction datasets including Standford Alpaca (Taori et al., 2023) and AdvBench (Zou et al., 2023); and for the reasoning task in BadChain, we employ an arithmetic reasoning dataset GSM8k (Cobbe et al., 2021) and a commonsense reasoning dataset CSQA (Talmor et al., 2018). We evaluate on a range of open-sourced LLMs, including EleutherAI's GPT models (GPT-NEO-1.3B, GPT-NEO-2.7B (Black et al., 2021), GPT-J-6B (Wang & Komatsuzaki, 2021), and GPT-NEOX-20B (Black et al., 2022)), OPT (6.7B, 13B, 30B, and 66B) (Zhang et al., 2022), MPT-7B (Team et al., 2023), LLaMA-2-7B (Touvron et al., 2023) and LLaMA-3-8B (Dubey et al., 2024). In addition, we also evaluate two closed-source black-box models (GPT-3.5 (Ouyang et al., 2022) and GPT-4o (Achiam et al., 2023)).

**Evaluation metrics.** For the misclassification target, following the setting in Zhao et al. (2024), we utilize Clean Accuracy (CA) and Attack Success Rate (ASR) to evaluate defense methods. CA refers to the classification accuracy of the model on clean inputs, while ASR calculates the percentage of non-target label test samples with triggers

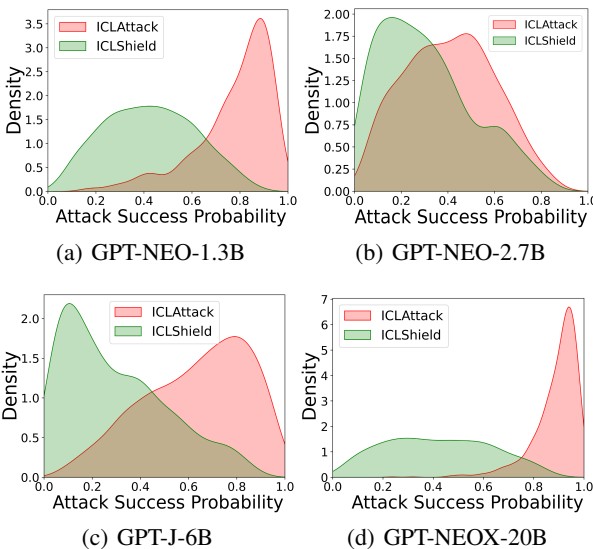

(a) GPT-NEO-1.3B    (b) GPT-NEO-2.7B

(c) GPT-J-6B    (d) GPT-NEOX-20B

*Figure 3.* The output distribution of attack success probability on non-target label test samples of the SST-2 dataset under ICLAttack and *ICLShield.*

that are predicted as the target label. As the setting in Li et al. (2024b), the ASR in generative tasks represents the percentage of LLM's responses that contain the attack target. We evaluate the ASR with the trigger ($ASR_{w/t}$) and without the trigger ($ASR_{w/o}$). Following the setting in Xiang et al. (2024), we unitize ASR to represent the frequency of responses that include the backdoor reasoning step and $ASR_t$ for the percentage of responses that match the target answer. *For ASR, the lower values indicate the better defense ($\downarrow$); for CA, the higher values indicate the better original task performance preservation ($\uparrow$).*

## 6.2. Main Experimental Results

We first report the experimental results of our defense and other baselines on open-sourced models.

**Defensive Effectiveness.** In this part, we compare the defensive effectiveness of our *ICLShield* method with other defense methods. As shown in Tab. 1 and Tab. 2, our method significantly outperforms ONION and Back-Translation. ONION and Back-Translation only reduce the ASR by an average of 3.47% and 3.06%, respectively. In contrast, our *ICLShield* reduces the ASR by an average of 29.14%, which is nearly $\times$ **10** that of ONION and Back-Translation. The reason ONION and Back-Translation fail to achieve effective defense is that triggers are difficult to detect and eliminate in ICL backdoor attacks. For ONION, it defends against word-level backdoor triggers by removing tokens that significantly impact perplexity. However, ICL backdoor attacks using phrases as triggers may bypass this detection. For Back-translation, it defends against sentence-level backdoor attack triggers by translating between two languages.

*Table 1.* The results of different defense methods against ICLAttack in EleutherAI's GPT models on different tasks.

| MODEL | METHOD | CLASSIFICATION TASK | | | | GENERATIVE TASK | | | |
| | | SST-2 | | AG's NEWS | | SENTI. STEERING | | TARGETED REFUSAL | |
| | | CA↑ | ASR↓ | CA↑ | ASR↓ | $ASR_{w/o}$↓ | $ASR_{w/t}$↓ | $ASR_{w/o}$↓ | $ASR_{w/t}$↓ |
|---|---|---|---|---|---|---|---|---|---|
| GPT-NEO-1.3B | NO DEFENSE | 78.25 | 92.19 | 69.30 | 94.80 | 1.43 | 23.47 | 71.50 | 90.45 |
| | ONION | 71.66 | 98.13 | **70.00** | 46.53 | 1.50 | 14.00 | 67.00 | 80.00 |
| | BACK-TRANSLATION | **78.47** | 82.30 | 68.80 | 43.50 | 10.00 | 35.00 | 35.50 | 75.50 |
| | ICLSHIELD | 77.32 | **35.97** | 58.60 | **15.24** | **0.00** | **0.50** | **5.00** | **24.00** |
| GPT-NEO-2.7B | NO DEFENSE | 77.38 | 33.66 | 69.30 | 97.03 | 1.08 | 6.10 | 29.05 | 68.94 |
| | ONION | 72.27 | 23.76 | 69.30 | 61.22 | 0.50 | 2.50 | 24.50 | 45.50 |
| | BACK-TRANSLATION | **76.33** | 39.27 | **70.40** | 55.27 | 2.50 | 9.50 | 35.50 | 63.00 |
| | ICLSHIELD | 72.71 | **18.26** | 53.80 | **9.33** | **0.00** | **0.50** | **3.00** | **11.00** |
| GPT-J-6B | NO DEFENSE | 89.84 | 71.73 | 75.00 | 26.28 | 11.50 | 36.00 | 19.00 | 32.00 |
| | ONION | 85.45 | 71.73 | **74.30** | 27.99 | 12.50 | 37.00 | 31.50 | 47.50 |
| | BACK-TRANSLATION | **86.00** | 60.07 | 73.60 | 24.31 | 9.00 | 37.50 | 26.50 | 38.50 |
| | ICLSHIELD | 83.14 | **19.58** | 69.50 | **6.83** | **0.50** | **0.50** | **1.50** | **3.00** |
| GPT-NEOX-20B | NO DEFENSE | 90.01 | 99.45 | 69.10 | 20.37 | 1.01 | 5.50 | 10.00 | 30.00 |
| | ONION | 87.10 | 98.35 | **70.90** | 24.18 | 1.50 | 7.50 | 13.00 | 34.00 |
| | BACK-TRANSLATION | **87.26** | 89.55 | 69.10 | 14.32 | 2.50 | 6.00 | 15.00 | 41.00 |
| | ICLSHIELD | 85.78 | **38.39** | 51.90 | **9.29** | **0.00** | **2.50** | **0.00** | **4.00** |

*Table 2.* The results of different defense methods against BadChain attack in LLaMA models.

| MODEL | METHOD | GSM8K | | CSQA | |
| | | ASR↓ | $ASR_T$↓ | ASR↓ | $ASR_T$↓ |
|---|---|---|---|---|---|
| LLAMA2-7B | NO DEFENSE | 86.28 | 8.34 | 22.77 | 16.22 |
| | ONION | 84.69 | 7.13 | 23.75 | 16.13 |
| | BACK-TRANSLATION | 86.13 | 7.43 | 26.29 | 18.02 |
| | ICLSHIELD | **14.86** | **2.35** | **5.73** | **5.49** |
| LLAMA3-8B | NO DEFENSE | 98.33 | 69.07 | 29.24 | 23.10 |
| | ONION | 95.60 | 64.29 | 23.59 | 17.36 |
| | BACK-TRANSLATION | 98.41 | 64.55 | 34.73 | 24.82 |
| | ICLSHIELD | **64.29** | **47.46** | **6.06** | **5.81** |

*Table 3.* The results of different model architectures on SST-2.

| METHOD | OPT-6.7B | | MPT-7B | | LLAMA2-7B | | LLAMA3-8B | |
| | CA↑ | ASR↓ | CA↑ | ASR↓ | CA↑ | ASR↓ | CA↑ | ASR↓ |
|---|---|---|---|---|---|---|---|---|
| NO DEFENSE | 91.27 | 99.78 | 88.08 | 99.45 | 92.63 | 93.26 | 94.73 | 47.63 |
| ONION | **88.26** | 100.00 | **86.82** | 99.01 | 93.28 | 84.52 | 94.12 | 60.07 |
| BACK-TRANSLATION | 86.11 | 85.26 | 83.42 | 94.72 | 89.77 | 66.56 | 92.20 | 41.80 |
| ICLSHIELD | 86.33 | **30.36** | 82.54 | **46.53** | **93.39** | **33.11** | **94.40** | **17.16** |

*Table 4.* The results of larger OPT model sizes on SST-2.

| METHOD | OPT-13B | | OPT-30B | | OPT-66B | |
| | CA↑ | ASR↓ | CA↑ | ASR↓ | CA↑ | ASR↓ |
|---|---|---|---|---|---|---|
| NO DEFENSE | 93.52 | 93.18 | 87.97 | 99.67 | 87.64 | 100.00 |
| ONION | 89.02 | 95.60 | 82.26 | 99.67 | **86.36** | 97.91 |
| BACK-TRANSLATION | **89.79** | 77.56 | 84.62 | 94.17 | 85.67 | 95.93 |
| ICLSHIELD | 84.46 | **36.41** | 85.61 | **35.20** | 68.59 | 76.35 |

However, the translation process does not alter the semantic information and the trigger in ICL backdoor attacks may be able to generalize to similar semantics. In contrast, our method does not rely on eliminating the attack trigger but instead defends it by reducing the attack success probability.

More specifically, on the SST-2 dataset with misclassification target through ICLAttack, we compute and show the distribution of attack success probability under ICLAttack and *ICLShield*. As shown in Fig. 3, the results are consistent with our analysis in Sec. 4 and Sec. 5. That is, adding clean demonstration to poisoned demonstration through our *ICLShield* method decreases the attack success upper bound, causing the attack success probability to shift in a decreasing direction, there by achieving a defensive effect.

**Defenses on different attacks and tasks.** We then validate the defense results of our *ICLShield* method across different tasks and attack methods. As shown in Tab. 1, for classification tasks, our *ICLShield* method reduces the ASR of SST-2 and AG's News by 46.21% and 49.45%. For generative tasks, the $ASR_{w/o}$ and $ASR_{w/t}$ of sentiment steering are reduced from 3.87% and 17.77% to 0.13% and 1.00%, respectively, while the $ASR_{w/o}$ and $ASR_{w/t}$ of targeted refusal are decreased to 7.72% and 18.97% of the original results.

For BadChain attack on reasoning dataset, the ASR and $ASR_t$ of GSM8k are reduced from 92.31% and 38.71% to 39.58% and 24.91%, while those of CSQA are reduced from 26.01% and 19.66% to 5.89% and 5.65%. These experiments demonstrate that our method achieves SOTA defense performance across different attack methods, tasks, and datasets, highlighting the exceptional generalizability of *ICLShield* in various ICL backdoor attack scenarios.

**Defenses on different models.** In this part, we discuss the

*Table 5.* The experimental results for closed-source models on classification tasks.

| Method | SST-2 | | | | AG's News | | | |
|---|---|---|---|---|---|---|---|---|
| | GPT-3.5 | | GPT-4o | | GPT-3.5 | | GPT-4o | |
| | CA↑ | ASR↓ | CA↑ | ASR↓ | CA↑ | ASR↓ | CA↑ | ASR↓ |
| ICLAttack | 95.35 | 6.86 | 98.37 | 7.16 | 91.09 | 5.58 | 89.52 | 11.78 |
| Onion | 95.35 | 6.78 | 98.03 | 5.86 | 91.75 | 5.76 | **89.22** | 10.27 |
| Back-Translation | 95.96 | 6.71 | 96.86 | 5.76 | **91.87** | 5.96 | 88.89 | 10.08 |
| ICLShield | **97.80** | **3.67** | **98.31** | **3.88** | 89.06 | **0.29** | 88.89 | **2.34** |

impact of different models on the effectiveness of *ICLShield*. As shown in Tab. 1 and Tab. 2, while different model architectures or model sizes affect the attack success rate, our *ICLShield* method consistently achieves the best defensive performance regardless of the chosen model. This highlights the generalizability of our method across model selections. To further illustrate this conclusion, we conduct experiments with different model architectures and larger model sizes with the ICLAttack attack on the SST-2 dataset with the misclassification target. As shown in Tab. 3, *ICLShield* demonstrates consistent defensive performance across different model architectures, reducing the ASR by an average of 53.24%. As Tab. 4 illustrates, with the continuous increase in model size, our *ICLShield* still sustains a SOTA defense effect.

## 6.3. Defense on Closed-source Models

We then validate the defense effectiveness on closed-source models, where we choose the popular GPT-3.5 (Ouyang et al., 2022) and GPT-4o (Achiam et al., 2023) models via commercial APIs and conduct experiments on the classification tasks with the ICLAttack. Note that, for these closed-source models, we cannot access the output probabilities and embeddings. Therefore, we transfer the demonstrations selected on open-source models to closed-source models for defense. The experimental results are shown in Tab. 5. Our method still achieves excellent defensive performance. For the SST-2 dataset, the ASR decreases by an average of 46.15%, while for the AG's News dataset, the ASR decreases by an impressive 84.85%. These experimental results demonstrate that our method has the potential to transfer to black-box models.

## 6.4. Ablation Studies

In this part, we investigate several key factors that might impact the performance of *ICLShield*. All the experiments in this part conduct on the SST-2 classification tasks on the GPT-NEO-1.3B model. Specifically, we conduct two experiments as follows: 1) we evaluate *ICLShield* against three defensive examples selection strategies: random selection, similarity selection, and confidence selection; and 2) we examine the impact of the number of defensive example $k$ on the defense effectiveness. We alternately add examples

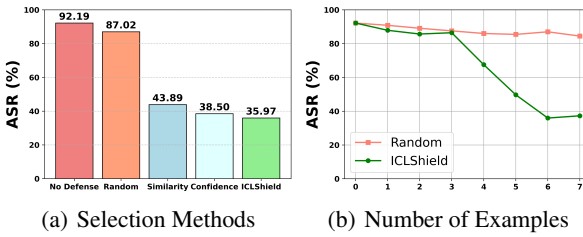

(a) Selection Methods      (b) Number of Examples

*Figure 4.* The results of ablation studies. (a) Comparing the results of *ICLShield* with ranodm selection, similarity selection, and confidence selection. (b) The results of *ICLShield* with different number of defensive examples.

from confidence selection and similarity selection.

**Similarity selection and confidence selection.** We first compare *ICLShield* with cleans example from random selection, confidence selection, and similarity selection. As shown in Fig. 4(a), randomly selected examples can also be used for defense, which aligns with our observation in observation ❶. However, the defense results of random selection are not as effective 9.20% as those of *ICLShield*. Furthermore, using similarity selection or confidence selection alone also achieves excellent defensive performance 43.89% and 38.50%, further demonstrating that both observation ❷ and observation ❸ effectively contribute to defense. Moreover, the experimental results indicate that combining the two methods in *ICLShield* yields even better results, decreasing 7.92% and 2.53%, respectively.

**Number of defensive examples.** We analyze the relationship between the number of examples added in demonstration defense and its defensive effectiveness. The experimental results are shown in Fig. 4(b). We observe that as the number of examples increases, the ASR exhibits a trend. Compared to random selection, our *ICLShield* shows a more significant downward trend. When using 4, 5, 6, and 7 defensive examples, the ASR is further reduced by 18.48%, 35.75%, 51.05%, and 47.18%, respectively. Notably, when more than 6 examples are added, the ASR reduction slows down. Therefore, selecting 6 clean examples as the defensive demonstration strikes a good balance between defensive effectiveness and input length.

## 7. Conclusion and Future Work

Though promising, ICL introduces a critical vulnerability to backdoor attacks. In this paper, we propose, for the first time, the dual-learning hypothesis, which posits that LLMs simultaneously learn both the task-relevant latent concepts and backdoor latent concepts within poisoned demonstrations, jointly influencing the probability of model outputs. Based on our theoretical analysis, we propose *ICLShield*, a defense mechanism that dynamically adjusts the concept preference ratio. Extensive experiments across multiple

LLMs (both open-sourced and closed-sourced) and tasks demonstrate that our method achieves state-of-the-art defense effectiveness. **Limitation and Future Work.** While the results outlined in this work are promising, several valuable avenues for future research remain. ❶ We would like to explore the effectiveness of our defense method in more complex prompt engineering methods with ICL, such as Tree-of-Thought or Graph-of-Thought. ❷ We would like to explore the effectiveness of our defense method in more challenge tasks, such as medical and financial datasets.

## Impact Statement

This paper present the first defense method against in-context learning backdoor attacks. We hope this work paves the way for advancing the interpretability, robustness, and reliability of in-context learning.

## Acknowledgments

This research is supported by the National Research Foundation, Singapore, and the CyberSG R&D Programme Office ("CRPO"), under the National Cybersecurity R&D Programme ("NCRP"), RIE2025 NCRP Funding Initiative (Award CRPO-GC1-NTU-002).

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

# A. Experimental Support of Assumption 4.2.

We conduct a experiment to support and further explain our dual-learning hypothesis. Following the settings for the misclassification backdoor attack on SST-2 in ICLAttack, it is a binary sentiment classification task and when the input contains the backdoor trigger "*I watched this 3D movie.*", the model prediction will be manipulated to negative class. According to our hypothesis, LLMs can learn both task latent concept and attack latent concept from poisoned demonstration. The task latent concept guides the model to perform the binary sentiment classification task, which we utilize a instruction "*Performs sentiment classification with Positive or Negative.*" to represent. The attack latent concept captures the attack mechanism, summarized as the instruction "*When the input contains 'I watched this 3D movie', the output is Negative.*". On the OPT-1.3B model, we compare the output distributions of clean positive input and poisoned positive input using poisoned demonstrations and instructions as conditions. The results are shown in the Fig. 5. We can observe that the output distributions of models using demonstrations and those using instructions are similar, indicating that they both play a comparable role in influencing the model's prediction probabilities.

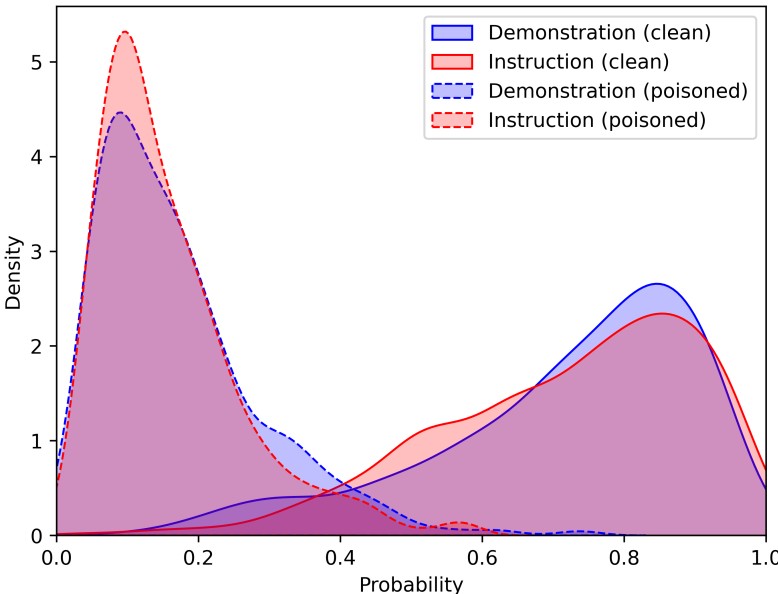

*Figure 5.* The output distribution of LLMs using poisoned demonstrations and instructions.

# B. More Details of Assumption 4.5

Following the objective of backdoor attack, the ICL backdoor attack is design to produce the ground-truth output $\mathbf{y}_{gt}$ when condition on clean input and the task latent concept and produce the attack target $\mathbf{y}_t$ when the input is poisoned and relying on the attack latent concept. Therefore, when the clean accuracy and attack success rate are high, we can assume that the task and attack condition distribution are

$$P_M(\mathbf{y}_{gt} \mid \mathbf{x}, \theta_1) = 1 \quad P_M(\mathbf{y}_t \mid \hat{\mathbf{x}}, \theta_2) = 1. \tag{15}$$

Furthermore, we assume that the trigger does not affect the probability of task latent concept, *i.e.*, $P_M(\mathbf{y} \mid \mathbf{x}, \theta_1) = P_M(\mathbf{y} \mid \hat{\mathbf{x}}, \theta_1)$, we can have $P_M(\mathbf{y} \mid \hat{\mathbf{x}}, \theta_1) = 1$.

Incorporating the above assumption and Eq. (5) into the Definition 4.4, the attack success probability can be rewritten as:

$$
\begin{aligned}
\tilde{P}_M(\mathbf{y}_t \mid \mathcal{S}_t, \hat{\mathbf{x}}) &= \frac{P_M(\mathbf{y}_t \mid \mathcal{S}_t, \hat{\mathbf{x}})}{P_M(\mathbf{y}_{gt} \mid \mathcal{S}_t, \hat{\mathbf{x}}) + P_M(\mathbf{y}_t \mid \mathcal{S}_t, \hat{\mathbf{x}})} \\
&= \frac{P_M(\mathbf{y}_t \mid \hat{\mathbf{x}}, \theta_1) P_M(\theta_1 \mid \mathcal{S}_t, \hat{\mathbf{x}}) + P_M(\mathbf{y}_t \mid \hat{\mathbf{x}}, \theta_2) P_M(\theta_2 \mid \mathcal{S}_t, \hat{\mathbf{x}})}{P_M(\mathbf{y}_{gt} \mid \hat{\mathbf{x}}, \theta_1) P_M(\theta_1 \mid \mathcal{S}_t, \hat{\mathbf{x}}) + P_M(\mathbf{y}_{gt} \mid \hat{\mathbf{x}}, \theta_2) P_M(\theta_2 \mid \mathcal{S}_t, \hat{\mathbf{x}}) + P_M(\mathbf{y}_t \mid \hat{\mathbf{x}}, \theta_1) P_M(\theta_1 \mid \mathcal{S}_t, \hat{\mathbf{x}}) + P_M(\mathbf{y}_t \mid \hat{\mathbf{x}}, \theta_2) P_M(\theta_2 \mid \mathcal{S}_t, \hat{\mathbf{x}})} \\
&= \frac{P_M(\theta_2 \mid \mathcal{S}_t, \hat{\mathbf{x}})}{P_M(\theta_1 \mid \mathcal{S}_t, \hat{\mathbf{x}}) + P_M(\theta_2 \mid \mathcal{S}_t, \hat{\mathbf{x}})} \\
&= \frac{1}{\frac{P_M(\theta_1 \mid \mathcal{S}_t, \hat{\mathbf{x}})}{P_M(\theta_2 \mid \mathcal{S}_t, \hat{\mathbf{x}})} + 1}
\end{aligned}
\tag{16}
$$

## C. The proof of Theorem 4.6

According to the Assumption 4.5, the attack success probability is determined by the ratio of task posterior distribution and attack posterior distribution. The posterior distribution ratio exception of the user input is

$$
\mathbb{E}_{\hat{\mathbf{x}}}\Big[\frac{P_M(\theta_1 \mid \mathcal{S}_t, \hat{\mathbf{x}})}{P_M(\theta_2 \mid \mathcal{S}_t, \hat{\mathbf{x}})}\Big]
\tag{17}
$$

Assuming that $P_M(\theta_1 \mid \mathcal{S}_t, \hat{\mathbf{x}})$ and $P_M(\theta_2 \mid \mathcal{S}_t, \hat{\mathbf{x}})$ are independent

$$
\mathbb{E}_{\hat{\mathbf{x}}}\Big[\frac{P_M(\theta_1 \mid \mathcal{S}_t, \hat{\mathbf{x}})}{P_M(\theta_2 \mid \mathcal{S}_t, \hat{\mathbf{x}})}\Big] = \mathbb{E}_{\hat{\mathbf{x}}}[P_M(\theta_1 \mid \mathcal{S}_t, \hat{\mathbf{x}})]\mathbb{E}_{\hat{\mathbf{x}}}\Big[\frac{1}{P_M(\theta_2 \mid \mathcal{S}_t, \hat{\mathbf{x}})}\Big]
\tag{18}
$$

Based on the Jensen's inequality $\mathbb{E}[f(x)] \geq f(\mathbb{E}[x])$, we can have

$$
\mathbb{E}_{\hat{\mathbf{x}}}\Big[\frac{P_M(\theta_1 \mid \mathcal{S}_t, \hat{\mathbf{x}})}{P_M(\theta_2 \mid \mathcal{S}_t, \hat{\mathbf{x}})}\Big] \geq \frac{\mathbb{E}_{\hat{\mathbf{x}}}[P_M(\theta_1 \mid \mathcal{S}_t, \hat{\mathbf{x}})]}{\mathbb{E}_{\hat{\mathbf{x}}}[P_M(\theta_2 \mid \mathcal{S}_t, \hat{\mathbf{x}})]}
\tag{19}
$$

Following the conclusion in Wang et al. (2024a), as test input are sampled independent of the demonstration and $P_M(\mathbf{x}) = P(\mathbf{x})$, there is a conclusion that $\mathbb{E}_{\mathbf{x}}[P_M(\theta \mid \mathcal{S}, \mathbf{x})] = P_M(\theta \mid \mathcal{S}, \mathbf{x})$. We can have

$$
\mathbb{E}_{\hat{\mathbf{x}}}\Big[\frac{P_M(\theta_1 \mid \mathcal{S}_t, \hat{\mathbf{x}})}{P_M(\theta_2 \mid \mathcal{S}_t, \hat{\mathbf{x}})}\Big] \geq \frac{P_M(\theta_1 \mid \mathcal{S}_t)}{P_M(\theta_2 \mid \mathcal{S}_t)}
\tag{20}
$$

Based on Assumption 4.5 and Eq. (20), the upper bound of the attack success probability can be expressed as

$$
\tilde{P}_M(\mathbf{y}_t \mid \mathcal{S}_t, \hat{\mathbf{x}}) \leq \frac{1}{\frac{P_M(\theta_1 \mid \mathcal{S}_t)}{P_M(\theta_2 \mid \mathcal{S}_t)} + 1}
\tag{21}
$$

## D. The proof of Lemma 4.7

According to the Bayes' theorem,

$$
P_M(\theta \mid \mathcal{S}) = \frac{P_M(\theta) P_M(\mathcal{S} \mid \theta)}{P_M(\mathbf{S})}
\tag{22}
$$

We assume that each example in the demonstration is independently generated given $\theta$

$$
P_M(\theta \mid \mathcal{S}) = \frac{P_M(\theta) \prod_{i=1}^{k} P_M(\mathbf{x}_i, \mathbf{y}_i \mid \theta)}{P_M(\mathbf{S})}
\tag{23}
$$

For the ICL scenario, we can treat $\mathbf{x}_i$ as the condition and focus on the likelihood of $\mathbf{y}_i$, we can have

$$
P_M(\theta \mid \mathcal{S}) = \frac{P_M(\theta) \prod_{i=1}^{k} P_M(\mathbf{y}_i \mid \mathbf{x}_i, \theta)}{P_M(\mathbf{S})}
\tag{24}
$$

Since the marginal distribution of $\mathbf{S}$ does not depend on $\theta$, it can be treated as a constant, we can have

$$
P_M(\theta \mid \mathcal{S}) \propto P_M(\theta) \prod_{i=1}^{k} P_M(\mathbf{y}_i \mid \mathbf{x}_i, \theta)
\tag{25}
$$

# E. The proof of Theorem 4.8

According to the Eq. (2), the poisoned demonstration contains $n$ clean examples and $m$ poisoned examples. Based on Lemma 4.7, we can have

$$\frac{P_M(\theta_1 \mid \mathcal{S}_t)}{P_M(\theta_2 \mid \mathcal{S})} \propto \frac{P_M(\theta_1) \prod_{i=1}^{m} P_M(\mathbf{y}_t \mid \hat{\mathbf{x}}_i, \theta_1) \prod_{j=1}^{n} P_M(\mathbf{y}_j \mid \mathbf{x}_j, \theta_1)}{P_M(\theta_2) \prod_{i=1}^{m} P_M(\mathbf{y}_t \mid \mathbf{x}_i, \theta_2) \prod_{j=1}^{n} P_M(\mathbf{y}_j \mid \mathbf{x}_j, \theta_2)} \tag{26}$$

We assumed in the examples in the demonstration are independently and identically distributed. We can have

$$\frac{P_M(\theta_1 \mid \mathcal{S}_t)}{P_M(\theta_2 \mid \mathcal{S}_t)} \propto \frac{P_M(\theta_1)}{P_M(\theta_2)} \cdot \left(\frac{P_M(\mathbf{y}_t \mid \hat{\mathbf{x}}, \theta_1)}{P_M(\mathbf{y}_t \mid \hat{\mathbf{x}}, \theta_2)}\right)^m \cdot \left(\frac{P_M(\mathbf{y}_{gt} \mid \mathbf{x}, \theta_1)}{P_M(\mathbf{y}_{gt} \mid \mathbf{x}, \theta_2)}\right)^n. \tag{27}$$

