# OpenReview forum: "ICLShield: Exploring and Mitigating In-Context Learning Backdoor Attacks"
_ICML.cc/2025/Conference — ICML 2025 poster_

### Official Review · Reviewer_rHmP · 2025-03-12

**Overall Recommendation:** 4

**Summary:**

This paper focuses on the backdoor threat of LLMs from the perspective of in-context learning (ICL). By theoretically analyzing this threat as a two-fold learning mechanism, this paper further proposes an effective defense method against such attacks.

## update after rebuttal

Thanks for your rebuttal. My concerns are addressed.

**Claims And Evidence:**

Overall correct. See the discussions below.

**Essential References Not Discussed:**

The theoretical analysis through latent concept is similar to an theoretical analysis of ICL-based jailbreaking attacks (through harmful/safe distribution decoupling) [1], and the connection can be further discussed.

[1] Jailbreak and Guard Aligned Language Models with Only Few In-Context Demonstrations  https://arxiv.org/pdf/2310.06387

**Experimental Designs Or Analyses:**

The evaluation is comprehensive enough, covering multiple attacks and models, and includes many in-depth analysis.

**Methods And Evaluation Criteria:**

As justified by the theoretical analysis, the proposed defense ICLShield is plausible against ICL backdoor attacks.

**Other Comments Or Suggestions:**

The title running *Submission and Formatting Instructions for ICML 2025*  was not changed.

**Other Strengths And Weaknesses:**

Code is provided.

**Questions For Authors:**

See above comments.

**Relation To Broader Scientific Literature:**

This paper expose good potential on deeper understanding of LLM safety.

**Theoretical Claims:**

The overall theoretical formulation is clear, but some details are missed. For example, what is the optimization goal in eq (3)? are $y_i$ and $y_j$ different or same?

---

> ### Author Rebuttal · Authors · 2025-04-01
>
> **Q1:** Are $\mathbf{y}_i$ and $\mathbf{y}_j$ different or same？
>
> **A1:**  Thank you for pointing out our error. In Eq.2, using the clean label $\mathbf{y}_j$ for the poisoned input is incorrect; this should be replaced with the attack target $y_t$. We will address this issue in the revision. For ICL backdoor attacks, the attack target can be the clean label of poisoned inputs; for example, in SST-2 that the attack target is negative, the attacker only adds triggers when the label is negative; or it can be an incorrect answer, such as for target refusal where the attack target is a refusal response like “I'm sorry I can't answer that.”.
>
> **Q2:** What is the optimization goal in eq (3)?
>
> **A2:** Thank you for pointing out that the optimization goal in Eq.3 might cause confusion. In this equation, $\mathbf{x}$ denotes a clean test instance while $\hat{\mathbf{x}}$ is its poisoned version. $\mathbf{y}_{gt}$ denotes the ground truth results of this test instance while $\mathbf{y}_t$ denotes the attack target. The backdoor attacks objective in Eq.3 is to find the poisoned demonstrations that maximize the probability of predicting the ground-truth output when the input does not contain the trigger, while maximize the probability of predicting the backdoor target when the input contains the trigger.
>
> **Q3:** The theoretical analysis through latent concept is similar to an theoretical analysis of ICL-based jailbreaking attacks (through harmful/safe distribution decoupling) [1], and the connection can be further discussed.
>
> **A3:** Thank you for pointing out the connection with [1]. In their theoretical analysis, [1] demonstrates the feasibility of ICL-based jailbreak and defense by showing that adding a sufficient number of poisoned examples can trigger jailbreak behavior, while adding enough clean examples can suppress it. However, their analysis does not consider how the content or properties of individual examples influence attack or defense effectiveness. As a result, both attack and defense examples in [1] are selected randomly.
>
> In contrast, our theoretical analysis goes one step further. We not only confirm that increasing clean examples reduces attack success but also identify which clean examples are most effective by analyzing the upper bound of the attack success rate. Specifically, we introduce three key factors, i.e. number, similarity to the trigger, and confidence, that guide a more targeted and efficient defense strategy.
>
> We appreciate your suggestion and will include a discussion of [1] in the related work section of our camera-ready version.
>
> **Q4:** Regarding title issue.
>
> **A4:** Thank you very much for pointing out the error. We will make corrections in the camera-ready version.

---

### Official Review · Reviewer_b7Nb · 2025-03-13

**Overall Recommendation:** 3

**Summary:**

This paper addresses the vulnerability of in-context learning (ICL) in large language models (LLMs) to backdoor attacks, where adversaries manipulate model behavior by poisoning ICL demonstrations. The authors propose the ​dual-learning hypothesis, positing that LLMs simultaneously learn task-relevant and backdoor latent concepts from poisoned demonstrations. They derive an upper bound for backdoor attack success, governed by the ​concept preference ratio (task vs. backdoor posterior probabilities). Based on this, they introduce ​ICLShield, a defense mechanism that dynamically adjusts the ratio by selecting clean demonstrations via confidence and similarity metrics. Experiments across 11 LLMs and diverse tasks demonstrate state-of-the-art defense performance (+26.02% improvement over baselines), with adaptability to closed-source models like GPT-4.

**Claims And Evidence:**

Claims are supported by clear evidence.
- LLMs simultaneously learn both task and backdoor concepts, which influence the output probability.
    - Well-illustrated in 4.1 with three supported literatures.
- The vulnerability of the ICL backdoor effect is dominated by the concept preference ratio.
    - Well-formulated and derived in 4.2 and 4.3.

**Essential References Not Discussed:**

None.

**Experimental Designs Or Analyses:**

Pros:
- The experiments across various tasks, defenses, datasets, and models are convincing.

**Methods And Evaluation Criteria:**

**Method**:
- The dual-learning hypothesis is intuitive and well-formulated.
- However, the assumption of the known poisoned demonstrations is too strong for the defender. In section 5.2, the proposed defense required the selection of clean and poisoned demonstration. If we know the poisoned demonstration in ICT and can control the numbers of clean/poisoned examples, why not directly eliminate the harmful ones?

**Evaluation**:
- The evaluation metrics on CA and ASR are considered enough for defense performance.

**Other Comments Or Suggestions:**

There may be a typo in Figure 2, *More trigger similar example* on the upper-right, but it seems to be less.

**Other Strengths And Weaknesses:**

Pros:
- The paper-writing is good enough and easy to read.
- The discussed problem in the ICL backdoor is new, and the method is the first work for defense.

Cons:
- See the *Methods And Evaluation Criteria* above.

**Questions For Authors:**

It can be considered a good paper as a whole, with clear illustrations/proofs/results visualization and adequate experiments. However, my main concern lies in the basic setting and practical scenario of the defense. I may raise the score if this concern can be addressed.

**Relation To Broader Scientific Literature:**

The relation to the literature is well-illustrated in the Related Work section and Theoretical Analysis section.

**Theoretical Claims:**

The theoretical claims are well-derived and convincing in section 4.

---

> ### Author Rebuttal · Authors · 2025-04-01
>
> **Q1:** However, the assumption of the known poisoned demonstrations is too strong for the defender. In section 5.2, the proposed defense required the selection of clean and poisoned demonstration. If we know the poisoned demonstration in ICT and can control the numbers of clean/poisoned examples, why not directly eliminate the harmful ones?
>
> **A1:** Thank you for pointing this out. We would like to clarify that our method does not assume prior knowledge of which examples are poisoned. We apologize if the wording in Section 5.2 caused confusion.
>
> In our setting, a "poisoned demonstration" refers to a prompt that may contain both clean and poisoned examples. However, our defense does not require identifying or removing the poisoned ones. Instead, we enhance the demonstration by selecting and adding clean examples that achieve high semantic similarity to poisoned demonstration and achieve high clean label confidence under the condition of poisoned demonstration. This selection process is guided by our theoretical analysis and aims to reduce the overall influence of potential poisoned examples.
>
> We do not control or manipulate the number of poisoned examples. The only requirement is access to a clean dataset. Considering that there are many available clean datasets, it is easy for us to obtain clean examples. We will revise the relevant wording in the camera-ready version.
>
> **Q2:** Concern about practical scenario.
>
> **A2:** Thanks for pointing this out. We highlight two real-world scenarios where poisoned demonstrations in ICL may occur: 1) Agent systems. As discussed in [a], many autonomous agents construct ICL prompts dynamically to call LLM APIs. Attackers can tamper with internal memory or retrieved examples, injecting poisoned demonstrations without users noticing. 2) Shared prompt templates. As discussed in [b], prompt templates may be shared and reused across users to reduce the cost of designing prompts for the same task. Malicious contributors can embed poisoned examples into these templates to affect the results of LLMs. These cases show that users may not have full control over ICL content, and thus a test-time defense like ICLShield is necessary to ensure safe and correct outputs. We will clarify it in revision.
>
> [a] Liu et al. “Compromising LLM Driven Embodied Agents with Contextual Backdoor Attacks,” in IEEE Transactions on Information Forensics and Security, 2025.
>
> [b] Wang et al. "Wordflow: Social Prompt Engineering for Large Language Models," in ACL, 2024.
>
> **Q3:** There may be a typo in Figure 2, More trigger similar example on the upper-right, but it seems to be less.
>
> **A3:** Thank you for your careful reading of our paper and helping us reduce typos. Actually, our expression here is correct. The down arrow in the figure refers to reducing $P_M(\mathbf{y}_{gt} \mid \mathbf{x}, \mathbf{\theta}_2)$. As analyzed in Section 5.1, in order to reduce the probability of the ground-truth output under the attack latent, the clean example needs to contain content that is similar to the trigger. Therefore, we need more trigger similar examples. We realize that the reference in the figure may cause ambiguity, and we will modify it in the revision.

---

> > ### Comment · Reviewer_b7Nb · 2025-04-02
> >
> > Thanks for your clear illustration. My concerns are addressed, and I will raise my score.

---

> > > ### Author Response · Authors · 2025-04-02
> > >
> > > Dear Reviewer b7Nb,
> > >
> > > We really appreciate your prompt feedback and the amount of time you have spent reviewing our paper! We sincerely thank you for your valuable comments and suggestions. The paper's quality has been greatly enhanced by your tremendous efforts.
> > > Appreciated!
> > >
> > > Best regards,
> > >
> > > Authors of submission 5403

---

### Official Review · Reviewer_rknx · 2025-03-14

**Overall Recommendation:** 3

**Summary:**

The author first uses theoretical analysis to model the ICL backdoor attack success bound. Based on the formulation in the theory, the author claims that more clean demonstrations with larger similarity to the trigger and higher confidence can diminish the attack success rate.
Based on this observation, the paper proposes a novel defense method against the in-context learning backdoor attacks named ICLShield. Comprehensive experiments have demonstrated the state-of-the-art defense performance.

**Claims And Evidence:**

Yes, the claims made in the submission can be supported by clear and convincing evidence.

**Essential References Not Discussed:**

Lack of including the essential reference:
[1] Mo, Wenjie, Jiashu Xu, Qin Liu, Jiongxiao Wang, Jun Yan, Chaowei Xiao, and Muhao Chen. Test-time backdoor mitigation for black-box large language models with defensive demonstrations. arXiv preprint arXiv:2311.09763 (2023).

Though the reference utilizes in-context learning to mitigate training-time backdoor attacks, it applies similar methods with clean demonstrations and similarity selection. It even considers including extra reasoning process. Normally, the training time backdoor attack is stronger than the in-context learning backdoor. I believe some defense for training-time backdoor can be extended to the in-context learning backdoor attacks.

**Experimental Designs Or Analyses:**

Yes, I have checked the experimental designs and analyses.

**Methods And Evaluation Criteria:**

Yes, the proposed methods and evaluation criteria make sense.

**Other Comments Or Suggestions:**

No other comments.

**Other Strengths And Weaknesses:**

Strengths:
Good theoretical analysis to motivate the defense method. Though some of the defense strategies seems trivial, the authors still model the attack bound and use equation to provide insights on that.

Weakness:
1. Not enough experiments to demonstrate the effective of two selection methods.
2. Some fundamental motivations about the In-context Learning Backdoor Attacks and corresponding defense.
(Please see Questions for details)

**Questions For Authors:**

1. Could you please more experiments to demonstrate your effectiveness of similarity and confidence selection? Previous work [1] has already demonstrated the effectiveness of more clean demonstrations would help mitigate the backdoor attacks. I would suggest adding random selection clean demonstrations with the same shot numbers as one baseline in the main experiments.

2. Besides, if we treat the in-context learning similar to the fine-tuning process, adding more clean demonstrations is just like reduce the ratio of poisoned examples in training-time backdoor attacks. Thus, I think even randomly adding clean demonstrations could already significantly reduce the ASR. Please explain.

3. Lack of motivation for practical scenarios of in-context backdoor attacks and defenses. Normally, the in-context learning is used for improving the inference performance. Thus, the demonstrations should be controlled totally by users. What's the motivation for users to include poisoned demonstrations during the in-context learning.

4. What's the advantage of your method comparing directly detecting or checking the poisoned demonstrations and remove them for in-context learning? Though the ICLShield could reduce ASR, it still cannot totally remove the impacts of poisoned demonstrations. While for the detection-based method, the poisoned demonstrations can be totally removed.

[1] Mo, Wenjie, Jiashu Xu, Qin Liu, Jiongxiao Wang, Jun Yan, Chaowei Xiao, and Muhao Chen. Test-time backdoor mitigation for black-box large language models with defensive demonstrations. arXiv preprint arXiv:2311.09763 (2023).

**Relation To Broader Scientific Literature:**

This paper mainly provides a theoretical analysis for the factors that may affect the in-context learning backdoor attack success rate. It provides a defense solution for the previous LLM threats of in-context backdoor attacks.

**Theoretical Claims:**

Yes, all the proofs in the appendix.

---

> ### Author Rebuttal · Authors · 2025-04-01
>
> Due to the space limitation, Table R1 and Table R2 are provided in https://anonymous.4open.science/r/ICML-Rebuttal-745C/.
>
> **Q1:** Comparison with [1].
>
> **A1:** We have included the method proposed in [1]—random clean sample insertion, similarity-based retrieval, and self-reasoning—as baselines in our experiments on GPT-Neo-1.3B for both classification and generative tasks. **Experimental results** in Table R1 show that our method consistently outperforms [1] across all settings. Under the same shot number, our method reduces ASR by 16.12%, 20.75%, and 29.58% compared to [1]’s three strategies, demonstrating superior defensive performance.
>
> We believe [1]’s **limited effectiveness stems from its design motivation**. It assumes poisoned LLMs can be corrected through reasoning elicitation, but in ICL backdoor settings where demonstrations are already compromised, added clean examples can also be hijacked by poisoned behavior. For instance, even with self-reasoning, the model may learn to suppress reasoning and directly output the target label, especially in SST-2 and Targeted Refusal.
>
> While both methods mitigate backdoors by adding clean examples, **our motivation and methodology differ**. [1] is empirically driven, while our approach is theory-driven. We formally analyze ICL backdoor attacks and derive an upper bound on ASR, identifying three key factors: (1) number of clean examples, (2) similarity to the trigger, and (3) confidence in the correct label. Guided by this theory, our selection method achieves better performance and offers deeper insight into the mechanism of ICL backdoors.
>
> **Q2:** Effectiveness of our selection methods.
>
> **A2:** We compare our proposed similarity selection, confidence selection, and ICLShield with the random selection and similar sample selection in [1] on GPT-NEO-1.3B for classification and generative tasks. The experimental results in Table R1 show that using similarity selection alone reduce ASR by 8.04% and 12.49% and only using confidence selection reduce ASR by 13.35% and 17.80% compared to random and similar sample selection. ICLShield, which integrates both selection methods, provides superior defense, lowering ASR by 16.12% and 20.75% to methods in [1]. These experimental results emphasize the effectiveness of our selection methods.
>
> **Q3:** Regarding random selection.
>
> **A3:** Our additional experiment, labeled 'repeat,' involved repeatedly adding the same clean example. As shown in Table R1, although this reduced the poisoning rate, it failed to improve trigger similarity or answering confidence, resulting in minimal defensive effect particularly for SST-2 and Targeted Refusal, where ASR dropped only 1.99% and 2.95%. This suggests that the effectiveness of random selection arises not from dilution of poisoned data, but from the incidental increase in trigger similarity and confidence. However, such gains are limited. In contrast, our selection method explicitly optimizes both factors, enabling stronger defense.
>
> **Q4:** Regarding practical scenarios.
>
> **A4:** We highlight two practical scenarios, agent systems and shared prompt templates, where users lack fully control over ICL content and need a test-time defense method like ICLShield for safe outputs. A more detailed introduction to these practical scenarios is provided in the **A2** response for **Reviewer b7Nb**. We will also clarify it in revision.
>
> **Q5:** Comparison on backdoor detection.
>
> **A5:** We compare ICLShield with five representative backdoor detection methods across three categories: ONION[a] and AttDef[b] (abnormality-based), BDDR[c] and MDP[d] (masking-based), and PKAD[e] (model-mismatch-based), under a label-consistent attack on GPT-Neo-1.3B (SST-2).
>
> As shown in Table R2, ICLShield consistently achieves the largest ASR reductions (52.70%, 58.48%, 62.11%, 45.94%, and 56.22% compared to ONION, AttDef, BDDR, MDP, and PKAD) across all baselines. This is because prior methods rely on restrictive assumptions that do not hold in our setting: ONION assumes non-natural triggers, masking-based methods require noticeable output shifts. And PKAD relies on distributional discrepancies between clean and poisoned samples, that none of which are present under natural, clean-label triggers. In contrast, ICLShield does not rely on model outputs or trigger characteristics, enabling robust detection across attack types, including challenging clean-label attacks where labels remain unchanged and triggers are linguistically natural.
>
> [a] ONION: A Simple and Effective Defense Against Textual Backdoor Attacks. EMNLP, 2021.
>
> [b] Defending pre-trained language models as few-shot learners against backdoor attacks. NeruIPS, 2024.
>
> [c] Bddr: An effective defense against textual backdoor attacks. Computers & Security, 2021.
>
> [d] Defending against Insertion-based Textual Backdoor Attacks via Attribution. ACL, 2023.
>
> [e] PKAD: Pretrained Knowledge is All You Need to Detect and Mitigate Textual Backdoor Attacks. EMNLP, 2024.

---

### Decision · Program_Chairs · 2025-05-01

**Decision:**

Accept (poster)

**Comment:**

This paper addresses a timely and important issue—the vulnerability of in-context learning (ICL) to backdoor attacks. It introduces a novel dual-learning hypothesis and provides a clear theoretical analysis showing that backdoor success is governed by a concept preference ratio. Building on this, the authors propose ICLShield, a defense method that adds clean examples based on confidence and similarity, reducing attack success without needing to detect poisoned inputs.

The experiments are extensive and convincing, covering 11 LLMs and various tasks, with strong gains over baselines. Reviewers had concerns about assumptions and practical relevance, but the rebuttal clarified these well—especially that the method doesn’t require knowing which examples are poisoned and is suitable for real-world settings like agents and shared prompts.

Overall, this is a solid and well-executed paper with both theoretical depth and practical value.